# Chemical Composition, Antioxidant Activities, Antidepressant Effect, and Lipid Peroxidation of Peruvian Blueberry: Molecular Docking Studies on Targets Involved in Oxidative Stress and Depression

**DOI:** 10.3390/plants13121643

**Published:** 2024-06-14

**Authors:** Iván M. Quispe-Díaz, Roberto O. Ybañez-Julca, Ricardo Pino-Ríos, José D. Quispe-Rodríguez, Daniel Asunción-Alvarez, Elena Mantilla-Rodríguez, Roger A. Rengifo-Penadillos, Edison Vásquez-Corales, Ricardo D. D. G. de Albuquerque, Wilfredo O. Gutiérrez-Alvarado, Julio Benites

**Affiliations:** 1Facultad de Farmacia y Bioquímica, Universidad Nacional de Trujillo, Trujillo 13011, Peru; iquispe@unitru.edu.pe (I.M.Q.-D.); jquispero@unitru.edu.pe (J.D.Q.-R.); hasuncion@unitru.edu.pe (D.A.-A.); amantilla@unitru.edu.pe (E.M.-R.); rrengifo@unitru.edu.pe (R.A.R.-P.); rgalhardod@unitru.edu.pe (R.D.D.G.d.A.); 2Laboratorio de Química Medicinal, Química y Farmacia, Facultad de Ciencias de la Salud, Universidad Arturo Prat, Casilla 121, Iquique 1100000, Chile; rpinorios@unap.cl; 3Instituto de Química Medicinal, Universidad Arturo Prat, Casilla 121, Iquique 1100000, Chile; 4Escuela de Farmacia y Bioquímica, Universidad Católica Los Ángeles de Chimbote, Chimbote 02801, Peru; evasquezc@uladech.edu.pe; 5Facultad de Farmacia y Bioquímica, Universidad Nacional de la Amazonía Peruana, Iquitos 16001, Peru; wilfredo.gutierrez@unapiquitos.edu.pe

**Keywords:** *Vaccinium corymbosum*, antioxidant activities, antidepressant effect, molecular docking, antioxidant enzyme, oxidative stress

## Abstract

Blueberries (*Vaccinium corymbosum* L.) are cultivated worldwide and are among the best dietary sources of bioactive compounds with beneficial health effects. This study aimed to investigate the components of Peruvian blueberry using high-performance liquid chromatography coupled to electrospray ionization and quadrupole time-of-flight mass spectrometry (HPLC–ESI–QTOF–MS/MS), identifying 11 compounds. Furthermore, we assessed in vitro the antioxidant activity and in vivo the antidepressant effect using a rat model and protective effect on lipid peroxidation (in the serum, brain, liver, and stomach). We also conducted molecular docking simulations with proteins involved in oxidative stress and depression for the identified compounds. Antioxidant activity was assessed by measuring total phenolic and flavonoid contents, as well as using 1,1-diphenyl-2-picrylhydrazin (DPPH), 2,2′-azino-bis-(3-ethylbenzothiazoline-6-sulfonic) acid (ABTS^•+^), and ferric-reducing antioxidant power (FRAP) assays. Peruvian blueberries demonstrated higher antioxidant activity than *Vaccinium corymbosum* fruits from Chile, Brazil, the United States, Turkey, Portugal, and China. The results showed that oral administration of Peruvian blueberries (10 and 20 mg/kg) for 28 days significantly (*p* < 0.001) increased swimming and reduced immobility in the forced swimming test (FST). Additionally, at doses of 40 and 80 mg/kg, oxidative stress was reduced in vivo (*p* < 0.001) by decreasing lipid peroxidation in brain, liver, stomach, and serum. Molecular docking and absorption, distribution, metabolism, excretion, and toxicity (ADMET) predictions were performed. In the molecular docking studies, quercitrin and 3,5-di-O-caffeoylquinic acid showed the best docking scores for nicotinamide adenine dinucleotide phosphate (NADPH) oxidase, superoxide dismutase, and xanthine oxidase; while 3,5-dicaffeoylquinic acid methyl ester and caffeoyl coumaroylquinic acid had the best docking scores for monoamine oxidase and serotonin receptor 5-HT_2_. In summary, our results suggest that the antidepressant and protective effects against lipid peroxidation might be related to the antioxidant activity of Peruvian *Vaccinium corymbosum* L.

## 1. Introduction

*Vaccinium corymbosum* L., a species within the genus *Vaccinium* and subgenus *Cyanococcus*, is part of the Ericaceae family [1]. This species is native to North America, where it thrives in a wide range of habitats including forests, marshes, and grasslands. In recent years, there has been a significant increase in blueberry production in the Southern Hemisphere, particularly in countries such as South Africa, China, Colombia, Chile, and Peru [2]. Peru, in particular, has emerged as a leading exporter of blueberries globally [3]. In Peru, blueberries are primarily cultivated in coastal deserts like the district of Viru in the La Libertad Region, where the temperate climatic conditions favor high-quality and early blueberry production.

The demand for *V. corymbosum* L. and its hybrids has surged over the past few decades due to market expansion [4]. Blueberries are recognized for their ability to provide natural antioxidants that greatly enhance their nutritional value. They are considered one of the top five healthiest fruits by the Food and Agriculture Organization (FAO) [5]. Consumers often regard blueberries as “superfoods”, “functional foods”, “nutrient powerhouses”, and “natural health packages” due to their rich content of micronutrients, polyphenols [6], and anthocyanins [7], all of which significantly contribute to human health. Blueberries are known to be rich in bioactive compounds that offer numerous health benefits, including antioxidant, anti-inflammatory, antiseptic, antiproliferative, antiaging, antibacterial, antidiabetic, astringent, and neuro-, cardio-, vision, and kidney protection properties [6,7,8].

Research has shown that extracts from various parts of the blueberry plant, including roots, stems, leaves, and fruits, contain several important compounds. These include anthocyanins (such as delphinidin, cyanidin, petunidin, peonidin, and malvidin, along with their O-glycosylated saccharides), flavonols (such as quercetin and syringetin along with their O-glycosides), catechins, proanthocyanins, and hydroxycinnamic acid derivatives. The concentration of these compounds in blueberries varies depending on several factors, such as genotype, environmental conditions, ripeness, and storage [8,9]. These compounds continue to attract significant scientific interest due to their well-known antioxidant capacity, which helps protect against oxidative stress damage [10]. Oxidative stress has been suggested to play a role in the development of a variety of diseases, including psychiatric disorders such as depression and anxiety [10]. Antioxidants from blueberries, such as anthocyanins, flavonoids, and phenolic acids, are known to interact with enzymes like nicotinamide adenine dinucleotide phosphate (NADPH) oxidase, superoxide dismutase (SOD), and xanthine oxidase [11]. However, excessive generation of ROS and the lack of an efficient antioxidant response trigger deleterious cellular signaling responses, damaging lipids, proteins, and DNA, causing inflammation, neurodegeneration, and neuronal death [12,13]. In the brain, phospholipids are particularly susceptible to peroxidation mediated by high ROS levels, favoring the production of oxidative stress markers such as malondialdehyde (MDA) [14].

Given the high prevalence of depression and the continuous search for effective treatments, polyphenols in blueberries have gained attention for their potential to reduce depressive symptoms by mitigating oxidative stress [15]. Currently, bioactive compounds are gaining momentum for their ability to prevent oxidative stress and psychological disorders, thereby promoting human health. In this context, numerous studies have explored the chemical profile [16,17,18,19,20,21,22,23,24,25,26,27,28,29,30,31,32] and the various biological activities of the *Vaccinium* genus [5,6,7,9,33,34,35,36]. 

Considering the beneficial properties found in blueberry fruits, our study aims to analyze the chemical composition of Peruvian *Vaccinium corymbosum* L., which has not been extensively researched and is exported worldwide. We intend to quantify its total phenolic and flavonoid contents and evaluate its in vitro antioxidant activity using DPPH, ABTS^•+^, and FRAP assays. Additionally, we assessed its antidepressant effects in vivo using a rat model, determined lipid peroxidation levels in different tissues, and performed molecular docking simulations with proteins involved in oxidative stress and depression. Finally, we predict the ADMET properties of the compounds found in Peruvian blueberries. 

## 2. Results 

### 2.1. Chemical Composition of Vaccinium corymbosum L. Fruit Extract

HPLC–ESI–QTOF–MS/MS was used to analyze a lyophilized sample of *Vaccinium corymbosum* L. fruit extract, and 11 compounds were identified as coumarin (hydroxymethyl coumarin), phenolic acids (chlorogenic acid, p-coumaroyl sinapoyl tartaric acid, 5-O-caffeoylshikimic acid, 3,5-di-O-caffeoylquinic acid, 3,5-dicaffeoylquinic acid methyl ester, caffeoyl coumaroylquinic acid), and flavonoids (myricetin-3-O-glucoside, quercetin-3-O-glucoside, quercetin-3-O-arabinoside, and quercitrin) (Table 1). The compounds’ spectra are provided in the Appendix A. The data showed that phenolic acids were the dominant class of compounds in Peruvian blueberry fruits, followed by flavonoids (Figure 1). 

### 2.2. Total Phenolic, Flavonoid Content and Antioxidant Capacity of V. corymbosum Fruit Extract

The total phenolic content (TPC) and total flavonoid content (TFC) of Peruvian blueberry were 3.86 ± 0.09 mg GAE/g extract and 0.93 ± 0.03 mg QE/g extract, respectively. Figure 2 shows the results of antioxidant assays such as DPPH, ABTS^•+^, and FRAP. The extract of *Vaccinium corymbosum* fruits reduced DPPH radical (IC_50_: 12.49 ± 0.45 mg/mL), as shown in Figure 2A. On the other hand, the ABTS^•+^ radical-scavenging assay showed higher activity of the extract (IC_50_: 18.76 ± 0.55 mg/mL) compared to dihydroascorbic acid (Figure 2C). The results of the FRAP assay (Figure 2E) showed that the concentration of *V. corymbosum* required for the Fe^3+^-TPTZ complex to interact and be reduced to an intense blue Fe^2+^-TPTZ by 50% (EC_50_) was 53.22 ± 1.45 mg/mL, compared to Trolox 1 mM (antioxidant standard). Figure 2B,D,F show that the highest concentrations of antioxidant activity of *V. corymbosum* were significantly (*p* < 0.001) higher compared to dihydroascorbic acid (0.5 mM), but lower compared to quercetin (1.0 mM).

### 2.3. Antidepressant Effect

Figure 3A shows an increased swimming time in the FST in rats orally administered with Peruvian blueberry fruit extract for 28 days. The results suggest that blueberry fruit extract at doses of 10 mg/kg (282.8 ± 7.3 s) and 20 mg/kg (278.0 ± 9.4 s) significantly increase the swimming time (*p* < 0.001) compared to the control group (112.5 ± 6.7 s). The two groups receiving the blueberry extract showed a longer swimming time (*p* < 0.001) compared to the fluoxetine-treated group (205.0 ± 4.0 s). Figure 3B shows that blueberry fruit extract (10 mg/kg and 20 mg/kg) did not affect climbing time compared to the control group, while fluoxetine-treated rats (90.8 ± 14.1 s) showed a significant effect (*p* < 0.001) against control (27.5 ± 5.3 s). Figure 3C shows that the duration of immobility in rats significantly decreased (*p* <0.001) when fluoxetine was administered. Peruvian *V. corymbosum* fruit extract decreased immobility time to 5.2 ± 1.9 s and 5.6 ± 3.4 s at doses of 10 and 20 mg/kg, respectively, compared to the control-treated group (160.0 ± 7.1 s). There were no significant differences (*p* > 0.05) in immobility time between rats treated with fluoxetine or Peruvian blueberry extract.

### 2.4. Lipid Peroxidation

Our results revealed that fluoxetine administered for 21 days at a dose of 20 mg/kg significantly increased MDA levels in the rat brain compared to the stomach, liver, or serum (Figure 4). Figure 4A shows that the administration of Peruvian *V. corymbosum* fruit extract was able to reduce the levels of MDA in the rat brain that had been treated with fluoxetine (80 mg/kg: 0.98 ± 0.14 nmol vs. control: 2.14 ± 0.18 nmol, *p* < 0.001). Similar results to those obtained in the brain were observed in the stomach, where *V. corymbosum* fruit extract significantly decreased the lipid peroxidation (*p* < 0.001) induced by fluoxetine (40 mg/kg: 1.16 ± 0.1 nmol and 80 mg/kg: 0.93 ± 0.1 nmol vs. control group: 1.84 ± 0.1 nmol; Figure 4B). The administration of *V. corymbosum* fruit extract showed a significant effect on both liver and serum lipid peroxidation. Figure 4C shows that the blueberry extract reduced the levels of MDA in the liver at doses of 40 mg/kg (0.11 ± 0.03 nmol) and 80 mg/kg (0.14 ± 0.06 nmol) compared to the control group (1.3 ± 0.22 nmol). Similarly, the serum levels were also reduced at the same doses, with values of 0.53 ± 0.14 nmol and 0.37 ± 0.06 nmol, respectively, compared to the control group (1.3 ± 0.14 nmol, *p* < 0.001; Figure 4D).

### 2.5. Molecular Docking

Table 2 shows the free binding energy values for the ligand–protein complexes with the 11 compounds from the Peruvian blueberry extract identified by HPLC–ESI–QTOF–MS/MS. The average values are color-coded in a red–yellow–green scheme, with red being the most stable binding energies and green the least stable. As shown in Table 2, the compounds exhibited potential inhibitory effects on NADPH oxidase and xanthine oxidase, which are related to their antioxidant activity. However, in terms of antidepressant activity, they inhibited monoamine oxidase and the serotonin receptor 5-HT_2_. In terms of antioxidant capacity, the results show that the ligands have a higher affinity for NADPH oxidase and xanthine oxidase, with values reaching up to −10.2 (compound **11**) and −9.9 kcal.mol^−1^ (compound **5**), respectively. For antidepressant effects, compounds **7** and **8** exhibited the strongest binding energies for monoamine oxidase and serotonin receptor 5-HT_2_, respectively. However, compounds **7**–**10** displayed very good binding energies for both proteins. 

The left side of Figure 5 illustrates the optimal location for the interaction between compound **11** and NADPH oxidase, whereas the right side shows a two-dimensional representation of the bonding interactions between the residues and the ligand. Note the formation of hydrogen bonds resulting from a significant number of hydrogen bond acceptor oxygens, van der Waals interactions, and π (aromatic)–σ interactions. These interactions are robust enough to compensate for unfavorable donor–donor interactions (as indicated in the legend of Figure 5). The data presented in Table 2 revealed a consistent preference for NADPH oxidase and xanthine oxidase, with compounds **5** and **11** showing a higher contribution to the potential antioxidant capacity of the extract.

Figure 6 shows the most favorable positions obtained for the compound **8**–monoamine oxidase complex. It is evident from the figure that van der Waals forces predominate in this complex, with hydrogen bonds and π–σ or π–alkyl interactions playing minor roles. Additionally, only one repulsive interaction was observed, resulting in a better binding energy value. As depicted in Table 2, the averages demonstrated that compounds **7** and **8** exhibited the most favorable antidepressant behavior among the compounds studied. It is important to note that compounds **4** and **10** also displayed reasonable values. The 2D representations of the ligand–protein interactions for the remaining systems are provided in Appendix A.

### 2.6. ADMET Profiles of Blueberry Fruit Extract

The development of new compounds, including pharmaceuticals, pesticides, and consumer products, requires a thorough evaluation of their physicochemical properties, pharmacokinetics, and other factors. Computer simulations offer valuable alternatives to experimental methods. This information is particularly useful for assessing the environmental and human health risks. Pharmacokinetic parameters and toxicity data were obtained using the pkCSM Online Tool (Table 3).

The results of the ADMET analysis, as displayed in Table 3, indicated that seven of the identified compounds (**1**, **2**, **4**, **6**, **9**, **10**, and **11**) had a molecular weight of less than 500 g/mol, which is vital for penetrability [37]. According to the data, all molecules from the blueberry fruit extract exhibited Caucasian colon adenocarcinoma (Caco-2) permeability values below 1.30, as well as intestinal absorption (50%) below that value, indicating that they would not be absorbed in the small intestine, except for compound **1** (71%). The transdermal efficacy of the compounds, as demonstrated by their skin permeability, ranged from −2.374 to −2.735 cm/h, suggesting that they would penetrate the skin properly. It should be noted that molecules would have difficulty penetrating the skin if their logKp value is greater than −2.5 cm/hour [38]. Compounds **1**, **2**, and **4** had unacceptable volume distribution (VDss) values, whereas the other compounds had values higher than −0.15. Penetration through the blood–brain barrier (BBB) is a critical factor for minimizing side effects and toxicity. It is noteworthy that all compounds had a log BB value of less than 0.3, which suggests their ability to cross the blood–brain barrier [39]. Available data indicate that the permeability of these compounds into the central nervous system (CNS) varies between −2.555 and −1.218. Regarding metabolism, 11 of the compounds were found to inhibit cytochrome P450 2D6 and cytochrome P450 3A4 and thus did not interfere with CYP450 biotransformation reactions. The excretion parameters are illustrated in terms of total clearance, showing that only compound **3** (p-coumaroyl sinapoyl tartaric acid) has a negative value (−0.025 log mL/min/kg) and positive values for the remaining compounds, indicating rapid excretion. Finally, the acute oral toxicity in rats (LD_50_) ranged from 2.032–3.072 mol/kg (equivalent to 357.73–1586.54 g/kg). According to OECD test methods [40], compounds with an LD_50_ greater than 5 g/kg are typically classified as exhibiting minimal acute toxicity. The hepatotoxicity descriptor suggested that all compounds were devoid of liver toxicity. In summary, the 11 compounds are consistent with drug-like structures allowing further drug development, but they differ in terms of intestinal absorption and volume distribution.

## 3. Discussion

Our study demonstrated, for the first time, the chemical composition of Peruvian blueberries (*V. corymbosum* L.), including their TPC and TFC as well as their in vitro antioxidant capabilities. Additionally, in a rat model, we also demonstrated that the lyophilized blueberry extract exerts an antidepressant effect, likely due to its ability to reduce lipid peroxidation in the brain as well as in the liver, stomach, and serum. 

The main phytochemical compounds described in the literature for *V. corymbosum* berries are anthocyanins [19]. Their extraction requires an acidified alcoholic solution, and for this reason, they were not detected in our study. Polyphenols found in Peruvian *V. corymbosum* fruits were identified according to their chromatographic behavior by HPLC-ESI-QTOF-MS/MS. The eleven compounds that were unequivocally or putatively identified corresponded to coumarin, phenolic acids, and flavonoids. Similar results were previously reported for the same species from Italy (flavonoids, phenolic acids, and iridoids) [19], the United States (hydroxycinnamic acids, flavan-3-ols, flavonols, and chalcone glycosides) [22], and Romania (phenolic acids and flavonoids) [20]. Furthermore, other berry species belonging to the genus *Vaccinium*, such as *V. floribundum* [32] and *V. myrtillus* [26,41], have been reported to contain several compounds, similar to those found in Peruvian *V. corymbosum*. These results indicate that the formation of bioactive compounds in fruits depends on many factors, such as geographical environment, genotype, season, ripeness, and storage [42,43,44,45].

Blueberry fruits are known to have a high polyphenol content, which are potent antioxidants that may have protective effects against diseases related to free radical production. Our findings indicate that the TPC and TFC values of blueberry fruits from the Viru district in Peru were higher than those reported in Chile [17], China [23,24], Brazil [46], the USA [47,48,49], Turkey [50], and Portugal [51]. Surprisingly, the phenolic compounds found in *V. corymbosum* were greater than those of other species of the genus *Vaccinium*, such as *V. myrtillus* from Brazil [41], *V. virgatum* from the USA and Brazil [52,53], and *V. floribundum* from Peru and Ecuador [29,30]. However, some varieties of *V. corymbosum* grown in China [24], Italy [26], Germany [42], and the USA [54] have been found to have TPC values higher than those reported in our study. 

The number of phenolic compounds found in the Peruvian blueberry may partly explain its interesting activity as an antioxidant and free radical scavenger. Contreras et al. [21] suggest that the in vitro antioxidant activity of cultured blueberry seedlings is mainly due to chlorogenic acid and myricetin, two compounds that we also reported in Peruvian blueberry fruits. Moreover, chlorogenic acid is widely distributed in berry crops and currently used as a natural antioxidant [55]. Its antioxidant activity is associated with the number of hydroxyl groups in its molecular structure [56]. Furthermore, flavonols such as myricetin show high antioxidant activity, which correlates with their structure [57]. 

Studies have demonstrated the relationship between phenolic compounds in food and neurodegenerative diseases, such as depression, due to their ability to traverse the blood–brain barrier [58]. Our findings reveal, for the first time, that Peruvian blueberry fruits exert an antidepressant-like effect in the forced swimming test (FST) at doses of 10 mg/kg and 20 mg/kg. Peruvian blueberry fruits increased swimming time and reduced immobility time in rats in the FST, without a significant change in climbing time (Figure 3). Since there have been no previous studies on the antidepressant effects of *V. corymbosum*, our results were compared to those reported for other species within this genus. In a mouse model of stress-induced unpredictable chronic mild depression, Kumar et al. [59] demonstrated that *V. myrtillus*, administered at doses of 125, 250, and 500 mg/kg, reduced immobility in the forced swim test, likely through the modulation of the nitric oxide pathway. Similarly, the hydroalcoholic extract of *V. ashei* leaves administered at a dose of 50 mg/kg mitigated unpredictable chronic mild stress that led to depression-like behavior in rats [60]. Oliveira et al. [61] showed that *V. virgatum* fruit extract administered at a dose of 200 mg/kg prevented the depressant effect in an experimental model of metabolic syndrome caused by a highly palatable diet. *V. bracteatum* leaf extract at doses of 100 and 200 mg/kg was reported to reverse chronic restraint stress-induced depression-like behavior in mice. From a mechanistic point of view, such an effect may be mediated by the regulation of the PKA/ERKs/Akt/mTOR/CREB signaling pathways [62] as well as the regulation of the hypothalamic–pituitary–adrenal axis and serotonin turnover systems [63]. According to Spohr et al. [10], the tail suspension test, a method for evaluating antidepressant effects, revealed that administering *V. virgatum* at doses of 100 and 200 mg/kg for 7 days was effective in preventing depressive behavior by decreasing the immobility of mice exposed to lipopolysaccharides (LPSs). These findings, together with those reported by Vega Custodio et al. [64], support that pretreatment with *V. virgatum* extract prevented LPS-induced depressive-like behavior in mice. Another study reported that acute administration of *V. myrtillus* (300 and 400 mg/kg, orally) to mice 1 h before behavioral analysis significantly reduced immobility compared to the control group [41]. These results suggest that the phytochemical composition of the Peruvian blueberry, which is rich in flavonoid compounds, may be responsible for its antidepressant-like effects. It is important to note that myricetin-3-O-glucoside, quercetin-3-O-glucoside, quercetin-3-O-arabinoside, and quercitrin have been reported in Peruvian blueberry fruits (Table 1). There is scientific evidence that quercetin, a flavonoid found in plant foods and herbal medicines, effectively modulates serotonergic activity by attenuating mitochondrial monoamine oxidase A (MAO-A) activity in the mouse brain, thereby producing antidepressant-like effects [65]. In addition, chlorogenic acid regulates learning, memory, and cognitive ability as well as anxiety, depression, and other posttraumatic stress disorder (PTSD)-like symptoms [66]. Furthermore, it is important to note that this antidepressant-like effect may also be related to other compounds not yet investigated, considering the polyphenol-rich composition of blueberry fruit. 

The effect of Peruvian blueberry extract on lipid peroxidation, specifically malondialdehyde (MDA) levels, was investigated in fluoxetine-treated rats. This selective serotonin reuptake inhibitor (SSRI), commonly prescribed for treating depression, has been reported to increase oxidative stress [67]. In the stomach, fluoxetine affects gastric contractility and interacts with muscarinic, α-adrenergic, and serotoninergic receptors and/or the ongoing reuptake/release of serotonin [67]. In addition, fluoxetine increased biochemical indicators of oxidative stress, including superoxide dismutase (SOD), catalase (CAT), and MDA, in the liver, suggesting an increase in ROS and subsequent oxidative damage [68]. Furthermore, fluoxetine induced neuronal death and ROS generation via a mechanism involving the influx of copper ions [69]. Likewise, fluoxetine has the potential to depress ATP levels in neurons, since it indirectly and nonspecifically affects electron transport and the activity of mitochondrial ATPase, which inhibits oxidative phosphorylation in the brain [70]. Chronic exposure to fluoxetine causes downregulation in the expression of genes involved in myelination, a process that shapes brain connectivity and may contribute to the remediation of symptoms of psychiatric disorders [71].

Interestingly, our results demonstrated that oral administration of Peruvian blueberry fruit extract at doses of 40 mg/kg and 80 mg/kg for 21 days after fluoxetine (20 mg/kg) treatment reduced MDA levels in the brain, stomach, liver, and serum of the rats (Figure 4). These results are consistent with those of similar studies conducted using the European blueberry (*V. myrtillus*) that show cytoprotective effects against oxidative damage in rat hepatocytes [72]. Supplementation with *V. corymbosum* may help reverse age-related and oxidative-stress-induced decline in brain function [68], reduce hepatic MDA levels, and increase glutathione (GSH), SOD, and glutathione peroxidase (GPx1) levels [73]. On the other hand, *V. corymbosum* decreased the pro-oxidant status as evidenced by a reduction in brain MDA levels and an increase in GSH levels and GSH-Px activities. Similarly, treatment with blueberries also reduces apoptosis and neurotoxicity by acting as an antioxidant (radical scavenger) in brain tissue [74]. Other *Vaccinium* species have also been reported to reduce MDA levels and increase antioxidant enzyme levels in the whole brain [59] or in specific regions, such as the striatum [10,53], hippocampus [10,53,61], and cerebral cortex [10,53,61,75,76]. 

Finally, molecular docking and ADMET predictions were performed to quantify and identify possible interactions between the Peruvian blueberry fruit components and a protein target, which may serve as a plausible mechanism correlated with (and possibly explaining) the observed in vitro antioxidant and in vivo antidepressant effect. In molecular docking studies, quercitrin and 3,5-di-O-caffeoylquinic acid showed the best docking scores for nicotinamide adenine dinucleotide phosphate (NADPH) oxidase, superoxide dismutase, and xanthine oxidase, whereas 3,5-dicaffeoylquinic acid methyl ester and caffeoyl coumaroylquinic acid had the best docking scores for monoamine oxidase and serotonin receptor 5-HT_2_. 

In summary, our study provides experimental evidence of Peruvian blueberries showing antioxidant and antidepressant properties and protective effects against lipid peroxidation in brain, liver, stomach, and serum. However, further analyses are needed to clarify whether the antidepressant effect and protective effects against lipid peroxidation in Peruvian fruits can be attributed solely to a single bioactive compound or to the complementary, synergistic, and/or additive effects of multiple phytochemicals. This phenomenon will depend on many factors, including geographical and environmental conditions.

## 4. Materials and Methods

### 4.1. Chemicals, Drugs, and Solvents

Chemicals, drugs and solvents were purchased from different companies, such as Aldrich (St. Louis, MO, USA), Merck (Peruana S. A, Ate, Lima, Peru), and Eli Lilly (Indianapolis, IN, USA), and were used as supplied.

### 4.2. Plant Material

*Vaccinium corymbosum* L. fruits were collected in January 2023 in the district of Viru, located at a latitude of 8°25′45.53″ S and 23 m above sea level in the province of Trujillo, Region of La Libertad, Peru (Figure 7). Following collection, the specimens were identified and verified by the “Herbarium Truxillense de la Facultad de Ciencias Biologicas de la Universidad Nacional de Trujillo”. A voucher sample, designated as accession number HUT 5320, was subsequently deposited in this herbarium.

### 4.3. Sample Preparation

The fruits of *Vaccinium corymbosum* L. were initially selected, washed with distilled water, frozen at −80 °C (Arctiko, Nashville, TN, USA), and lyophilized with a freeze-dryer (Labconco, Kansas City, MO, USA). The Lyophilized samples were stored under protection at +4 °C for analysis and further tests.

### 4.4. Chemical Identification by HPLC-ESI-QTOF-MS/MS

Samples were prepared in methanol at a concentration of 1 mg/mL. The chromatographic separation was performed using a UPLC Rapid Resolution (WATERS/MICROMASS, Milford, MA, USA) composed of a binary pump, degreaser, and automatic injector and using a ZORBAX Eclipse Plus C-18 column (2.1 × 150 mm × 5 μm) with a flow rate of 0.4 mL/min and 2 μL injection. The elution gradient was acetonitrile (B) and water with 0.1% acetic acid (A) in the following ratio: 0.0–4.7 min 3–30% B; 4.7–7.8 min 30–50% B; 7.8–11 min 50–90% B; 11–12.5 min 90%; 12.5–14 min 90–93%. The column effluent was divided by a T-valve and a fraction equivalent to 20 μL/min was incorporated into the mass spectrometer. The chemical identification was performed using a Q-TOF orthogonal mass spectrometer (micrOTOF-QTM, Bruker Daltonics, Madison, WI, USA) equipped with an electrospray ionization (ESI) source. The analysis parameters were provided for the positive mode, with a mass range of 100–1000 *m*/*z*:4500 V capillary voltage, set end-plate offset −500 V, set charging voltage 2000 V, drying gas temperature 200 °C, drying gas flow 10.0 mL/min, gas pressure 4 bar, collision energy (MS/MS) 50 eV, and collision gas N_2_. The mass data obtained were processed using the Bruker Compass Data Analysis 4.2 software (Bruker Daltonics).

### 4.5. Total Phenolic Content (TPC)

The total phenolic content of the lyophilized sample of Peruvian *V. corymbosum* L. fruit extract was determined using the modified Folin–Ciocalteu method, as described by Singleton and Rossi [77]. A dilution of the sample (50 μL) was mixed with 450 μL of distilled water and 2.5 μL of Folin–Ciocalteu solution. After 5 min at 40 °C, 2 μL of 7% sodium carbonate was added, and the mixture was incubated at room temperature for 90 min. The resulting deep blue color was measured at 760 nm, and the results are expressed as mg of gallic acid equivalent (GAE)/mL of sample.

### 4.6. Total Flavonoid Content (TFC)

The measurement of the total flavonoid content was carried out using the aluminum chloride (AlCl_3_) colorimetric method, as described in [78]. A sample volume of 30 μL was mixed with 10 μL of 10% AlCl_3_, 10 μL sodium acetate, and 250 μL distilled water. After 30 min, the mixture was incubated (25 °C) with agitation. A yellow coloration was observed, and the absorbance was measured at a wavelength of 414 nm. The results are reported as the amount of quercetin equivalent (QE)/g of sample.

### 4.7. Antioxidant Capacity Assays

#### 4.7.1. FRAP Assay

The FRAP assay was performed according to a previously described method [79], with certain modifications. Briefly, the FRAP stock solutions included 300 mM acetate buffer (pH 3.4), 10 mM 2,4,6-tripyridyl-S-triazine (TPTZ) solution in 40 mM HCl, and 20 mM FeCl_3_ × 6H_2_O. The working solution was prepared by mixing 25 mL of acetate buffer, 2.5 mL of TPTZ solution, and 2.5 mL of FeCl_3_ × 6H_2_O solution, which was then warmed to 37 °C prior to use. A lyophilized sample of Peruvian *V. corymbosum* fruits (10 μL) was allowed to react with 300 μL of fresh FRAP solution for 60 min in the dark. The absorbance of the colored ferrous tripyridyltriazine complex was measured at 593 nm (*n* = 3). A standard curve was plotted using the standard antioxidant Trolox^®^.

#### 4.7.2. ABTS^•+^ Free-Radical-Scavenging Activity

The ABTS^•+^ radical-scavenging assay was carried out according to the described method [79] with some modifications. Specifically, the ABTS reagent was diluted with distilled water to a concentration of 7 mM. The ABTS^•+^ radical cation was produced by reacting the ABTS stock solution with 2.45 mM potassium persulfate (K_2_S_2_O_8_) in a volume ratio of 1:1, followed by incubation in the dark at room temperature for at least 12–16 h. The ABTS stock solution was then diluted with ethanol to obtain an absorbance of 0.710 ± 0.050 at 734 nm. Subsequently, 10 μL of the diluted sample was mixed with 260 μL of the ABTS^•+^ radical cation solution, and the absorbance of the reaction mixture was measured after 4 min at 734 nm using a Fisherbrand accuSkan GO UV/Vis Microplate Spectrophotometer (Hampton, VA, USA). A curve of % ABTS^•+^ radical versus concentration was plotted, and the IC_50_ values were calculated. IC_50_ represents the concentration of the sample required to scavenge 50% of the ABTS^•+^ radical cation.

#### 4.7.3. DPPH Free-Radical-Scavenging Activity

The antioxidant activity of the sample was evaluated using a stable free radical, 2,2-diphenyl-1-picrylhydrazyl (DPPH), according to a slightly modified method described in the literature [79]. With 25 μL of each dilution of the sample and 300 μL of a DPPH radical solution (0.1 mM), the reaction mixture was incubated for 30 min at room temperature. The absorbance was then measured at 517 nm using a Fisherbrand accuSkan GO UV/Vis Microplate Spectrophotometer (Hampton, VA, USA). A curve of the percentage of DPPH bleaching activity versus concentration was then plotted, and the IC_50_ values were calculated. IC_50_ denotes the concentration of the sample required to scavenge 50% of DPPH free radicals.

All assays were performed in triplicate and are reported as the mean values ± SD.

### 4.8. Animals

The experiments in this study were performed in accordance with the protocols of the American Veterinary Medical Association (AVMA) and the Ethics Committee of the Pharmacy and Biochemistry Faculty of the National University of Trujillo (COD.N°: P 012-19/CEIFYB). Sixty 8–10-week-old *Rattus norvegicus* Holtzman rats (weighing 200–250 g) were kept in cages at a temperature of 22–25 °C with a 12 h light/dark cycle. These animals were provided with unrestricted access to standard rat chow (Molinorte S.A.C., Trujillo, Peru) and water. A preliminary evaluation of the FST was performed on Peruvian blueberry at doses ranging from 10 to 80 mg/kg. The lowest doses (10 mg/kg and 20 mg/kg) that showed the most promising antidepressant results were selected for this study.

### 4.9. Forced Swimming Test (FST)

The apparatus comprised a transparent Plexiglas cylinder (height: 50 cm, diameter: 20 cm) filled with water (temperature: 24 ± 1 °C) to a depth of 30 cm. Prior to the 5 min swimming test, each animal was placed in the cylinder for a 15 min period, 24 h in advance. Peruvian *V. corymbosum* was administered to the animals at doses of 10 and 20 mg/kg on a daily basis for 28 days preceding the swimming test. During the 5 min swimming test, the following behaviors were observed: (a) swimming, which indicates movement through the swim chamber, including crossing into another quadrant; (b) climbing or thrashing, which refers to upward-directed movements of the forepaws along the side of the swim chamber; and (c) immobility, which is when the rat does not try to escape except for the necessary movements to keep its head above the water [80].

### 4.10. Lipid Peroxidation Model Chronically Treated with Fluoxetine and V. corymbosum in Different Rat Tissues

Twenty-four Holtzman rats were randomly divided into four experimental groups as follows: group 1 (*n* = 6; vehicle) consists of rats treated with NaCl 0.9% for 42 days, while group 2 (*n* = 6; Control “FLX”) consists of rats treated with fluoxetine (20 mg/kg) for 21 days, followed by NaCl 0.9% for an additional 21 days more. Group 3 (*n* = 6; Problem I “FLX + blueberry 40 mg/kg”) was composed of rats that were given fluoxetine (20 mg/kg) for 21 days, followed by *Vaccinium corymbosum* L. in doses of 40 mg/kg (for an additional 21 days). Group 4 (*n* = 6; Problem II “FLX + blueberry 80 mg/kg”) was composed of rats that were administered fluoxetine (20 mg/kg) for 21 days, followed by *Vaccinium corymbosum* L. at a dose of 80 mg/kg for an additional 21 days. At the end of the experiment, all groups of rats were sacrificed, and samples were collected in order to measure lipid peroxidation.

#### 4.10.1. Lipid Peroxidation in Serum

Blood was collected by cardiac puncture into a tube without anticoagulant, as previously described [81], and the sample was immediately centrifuged at 3500 rpm for 10 min. From the resulting centrifuged sample, 100 µL was mixed with 100 µL of butylated hydroxytoluene, 100 µL of FeCl_3_ × 6H_2_O, 150 µL of glycocola-HCl buffer (pH 3.5), and 150 µL of TBA. The mixture was incubated for 60 min at 5 °C and then heated in a water bath at 100 °C for another 60 min. Afterwards, 2.5 mL of a solution (15:1, *v*/*v*) of n-butanol-pyridine and distilled water (0.5 mL) was added, and the tubes were cooled in an ice bath and centrifuged at 4000 rpm for 10 min. The absorbance of the resulting supernatant was then measured at 532 nm.

#### 4.10.2. Lipid Peroxidation in Brain

MDA levels were evaluated utilizing a previously outlined method [80,82]. The brains were stored at 4 °C and homogenized in ice-cold potassium phosphate buffer (20 mM, pH 7.4) containing 140 mM KCl. The resulting mixture was then centrifuged at 1700× *g* for 10 min at 4 °C, and the supernatant was collected and incubated in a water bath at 37 °C in the dark for 60 min. Subsequently, 20 µL of 10% *w*/*v* TCA was added, and the mixture was centrifuged at 960× *g* for 10 min at 4 °C. Then, 1 mL of the supernatant was taken, and 2 mL of TBA (0.67%, *w*/*v*) was added. The resulting mixture was incubated at 100 °C for 60 min before being cooled on ice for 15 min. Next, 4 mL of n-butanol/pyridine (15:1, *v*/*v*) and 0.5 mL of distilled water were added. The supernatant was then centrifuged at 960× *g* for 10 min at 4 °C, and the absorbance was measured at 532 nm.

#### 4.10.3. Lipid Peroxidation in Liver 

The liver was perfused with a cold saline (0.9% NaCl) solution via the portal vein before being homogenized. The sample, weighing 2 g, was crushed and homogenized with 50.0 mM phosphate buffer at a pH of 7.4. The homogenate was then centrifuged at 10,000 rpm for 10 min at 4 °C. Subsequently, 50 µL of phosphate buffer (50 mM, pH 7.4), 1 mL of 10% (*w*/*v*) TCA, and 450 µL of the supernatant were transferred to a new tube and centrifuged at the same speed and temperature. To measure malondialdehyde levels, 1 mL of TBA and 1 mL of the supernatant were mixed and heated in a boiling water bath for 30 min. After cooling, 4 mL of n-butanol and pyridine (15:1, *v*/*v*) were added, and the values were measured using a spectrophotometer at a wavelength of 532 nm [80,82].

#### 4.10.4. Lipid Peroxidation in Stomach

The rat stomachs were removed and cleansed with cold saline (4 °C) three times. The corpus mucosa was then scraped, weighed, and homogenized in a solution consisting of 10 mL of 100 mM KCl and 0.3 mM EDTA (10%). A 0.5 mL aliquot of the homogenate was mixed with a solution containing 0.5 mL of Texapon^®^ and 1.5 mL of trichloroacetic acid (10%, *w*/*v*). The mixture was centrifuged at 3000 rpm for 10 min, and the resulting supernatant was mixed with 1.5 mL of thiobarbituric acid (0.8%; *w*/*v*). The mixture was then incubated at 98 °C for 1 h before being cooled, and the absorbance of the supernatant was measured at 532 nm. A standard curve was established using 1,1,3,3-tetramethoxypropane [83].

### 4.11. Computational Methods

Molecular docking studies with proteins involved in oxidative stress and depression were performed on compounds **1**–**11** of Peruvian *V. corymbosum* L. fruit extract. The first group of proteins evaluated for antioxidant capacity included NADPH oxidase [84] (PDB ID: 2CDU), superoxide dismutase [85] (PDI ID: 4MCM), and xanthine oxidase [86] (PDB ID: 3NRZ). The second group was used to evaluate the antidepressant capability of the proteins and included serotonin receptor 5-HT_2_ [87] (PDB ID: 6A94), monoamine oxidase [88] (PDB ID: 1O5W), and sodium/potassium-transporting ATPase alpha-1 chain [89] (PDB ID: 1MO7). The data for this analysis were obtained from the Protein Data Bank [90]. The molecular mechanics optimization was performed using the universal force field [91] (UFF) implemented in the Avogadro program [92], and the corresponding SMILES [93] can be found in the Appendix A. The proteins were prepared using the UCSF CHIMERA 1.17.3 software [94], and the docking calculations were carried out through the CB-DOCK2 server using a fully automated blind docking approach [95,96]. The binding sites were identified using the CurPocket method [97,98], which employs a protein-surface curvature-based cavity detection approach. The free binding energies were calculated using the AutodockVina program [99]. Results were compared with available literature data [100,101,102,103,104,105,106] to ensure the binding sites were appropriate for our study. The visualization was performed using the Biovia Discovery Studio visualizer.

### 4.12. ADMET Prediction

The pkCSM online tool (http://biosig.lab.uq.edu.au/pkcsm/prediction, accessed on 28 December 2023) [107] was used to predict absorption, distribution, metabolism, excretion, and toxicity (ADMET) of compounds **1**–**11** of Peruvian *V. corymbosum* fruit extract.

### 4.13. Statistical Analysis

The GraphPad Prism 8.0.2 software (San Diego, CA, USA) was utilized for the analysis. The variable *n* represents the number of animals that were studied, and the results are expressed as the mean ± standard deviation (SD) or standard error of the mean (SEM). To analyze the groups, a one-way or two-way analysis of variance (ANOVA) was conducted, followed by a Tukey post hoc test. A value of *p* < 0.05 was considered statistically significant. 

## 5. Conclusions

In conclusion, this study identified 11 compounds in the lyophilized Peruvian *Vaccinium corymbosum* fruit extract, including coumarins, phenolic acids, and flavonoids. A high antioxidant activity of the lyophilized extract of the fruit of Peruvian blueberry was found and it is suggested that it might be related to the results shown in the antidepressant and protective effects against lipid peroxidation. In addition, in the molecular docking studies, quercitrin and 3,5-di-O-caffeoylquinic acid had the best docking scores for NADPH oxidase, xanthine oxidase, and monoamine oxidase; while 3,5-dicaffeoylquinic acid methyl ester and caffeoyl coumaroylquinic acid had the best docking scores for monoamine oxidase and serotonin receptor 5-HT_2_. Finally, according to the ADMET prediction using the pkCSM online tool, most of the compounds displayed suitable pharmacokinetic properties, as shown by absorption, distribution, metabolism, excretion parameters, and low toxicities.

## Figures and Tables

**Figure 1 plants-13-01643-f001:**
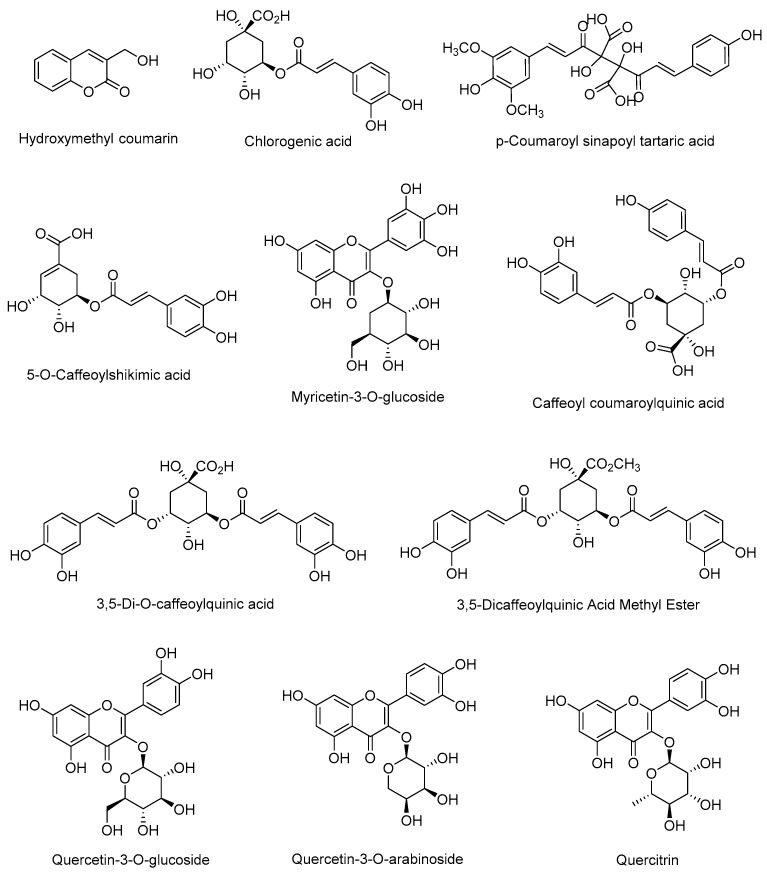
Structures of compounds from Peruvian *Vaccinium corymbosum* L. fruit extracts [19].

**Figure 2 plants-13-01643-f002:**
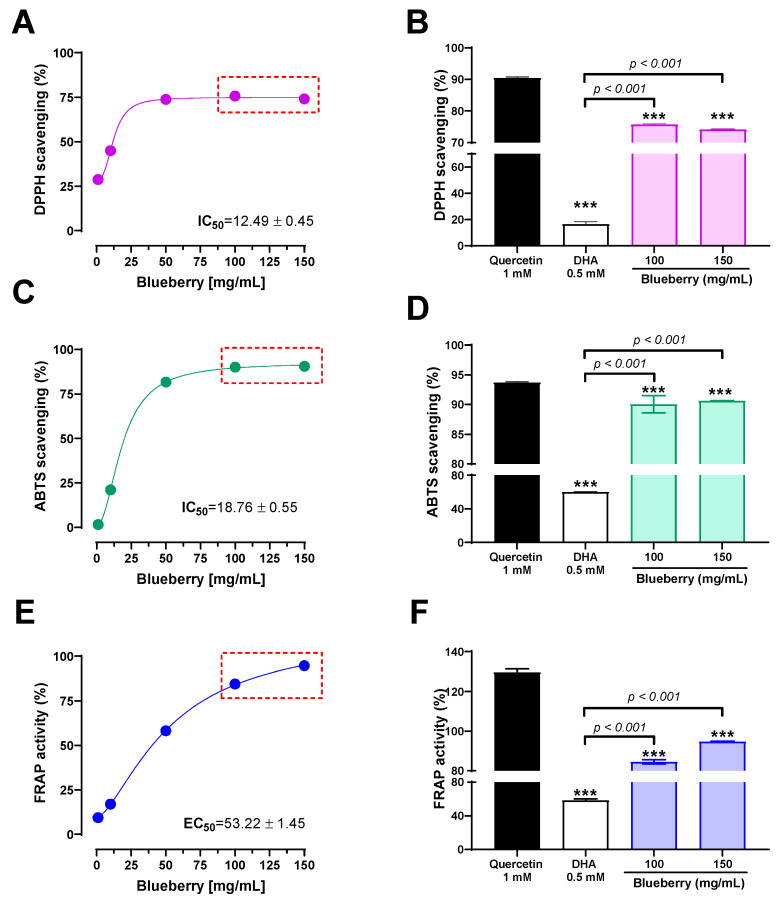
Antioxidant activity of Peruvian *Vaccinium corymbosum* L. by DPPH (**A**), ABTS^●+^ (**C**), and FRAP (**E**) assays. The red dashed lines indicate the comparison (panels **B**,**D**,**F**) between the highest concentrations of *Vaccinium corymbosum* and two positive controls, quercetin (1 mM) and dihydroascorbic acid (DHA, 0.5 mM). DPPH = 2,2-diphenyl-1-picrylhydrazyl radical; ABTS^●+^ = 2,20-azinobis (3-ethylbenzothiazoline-6-sulfonic acid); FRAP = ferric-reducing antioxidant power. Results are represented as the mean ± standard deviation (SD) of at least three experiments (*n* = 3–6). Significant differences: *** *p* < 0.001 vs. quercetin (1 mM).

**Figure 3 plants-13-01643-f003:**
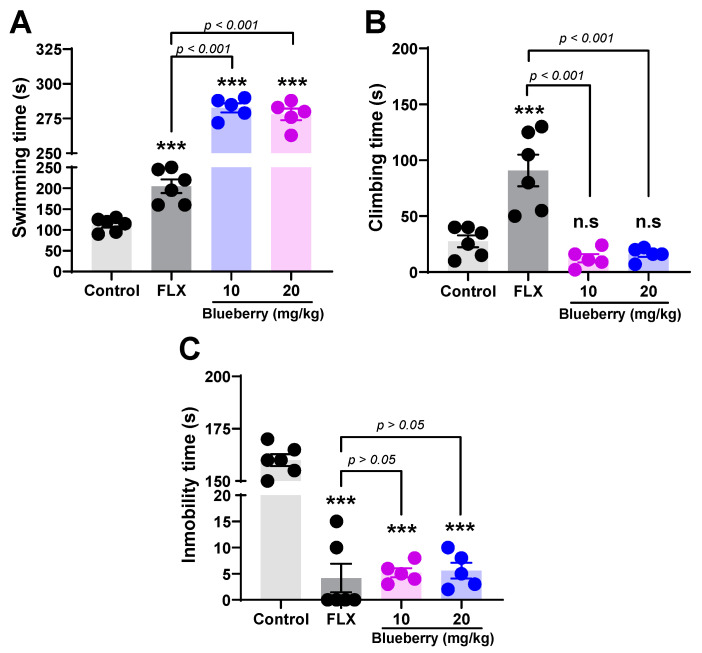
Effect of oral administration of Peruvian blueberry fruit extract and fluoxetine 20 mg/kg (FLX) in the forced swimming test. (**A**) Swimming time (s), (**B**) climbing time (s), and (**C**) immobility time (s). Results are presented as the mean ± SEM (*n* = 5–6). Comparisons test: *** *p* < 0.001 versus fluoxetine. ns: no significant difference vs. control group.

**Figure 4 plants-13-01643-f004:**
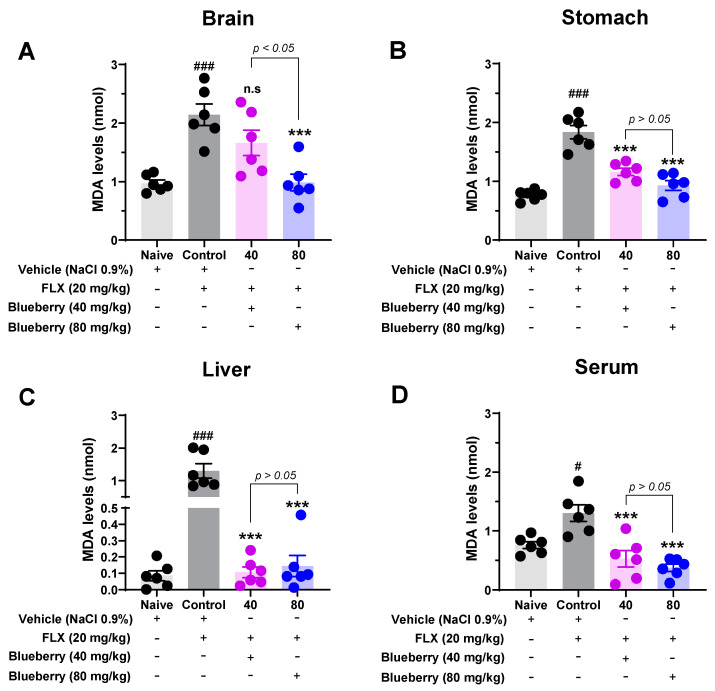
Effect of oral administration of blueberry fruit extract and fluoxetine (FLX) on oxidative stress in brain (**A**), stomach (**B**), liver (**C**), and serum (**D**). Results are presented as the mean ± SEM. Comparisons test: *** *p* < 0.001 versus control group; ^#^ *p* < 0.05, ^###^ *p* < 0.001 vs. naive group (vehicle). ns: no significant difference vs. control group.

**Figure 5 plants-13-01643-f005:**
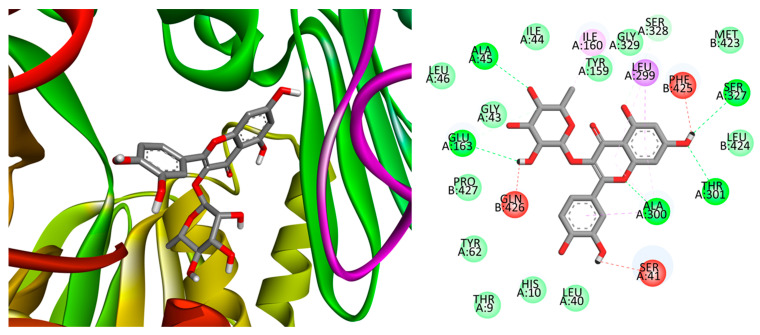
Best position obtained for the compound **11**–NADPH oxidase complex (**left**) and 2D representation of the interactions between the ligand and residues (**right**). Hydrogen atoms are omitted for clarity in some cases. Green = conventional hydrogen bond; light green: van der Waals interactions; red = unfavorable donor–donor interaction; purple = π (aromatic)–σ interactions; and pink = alkyl– or π–alkyl interactions.

**Figure 6 plants-13-01643-f006:**
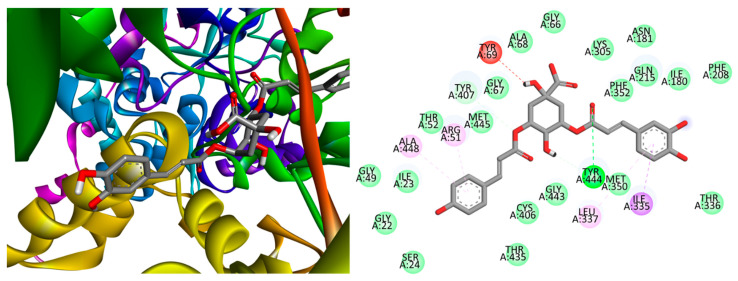
Best position obtained for the compound **8**–monoamine oxidase complex (**left**) and 2D representation of the interactions between the ligand and residues (**right**). Hydrogen atoms are omitted for clarity in some cases. Green = conventional hydrogen bond; light green: van der Waals interactions; red = unfavorable donor–donor interaction; purple = π (aromatic)–σ interactions; and pink = alkyl– or π–alkyl interactions.

**Figure 7 plants-13-01643-f007:**
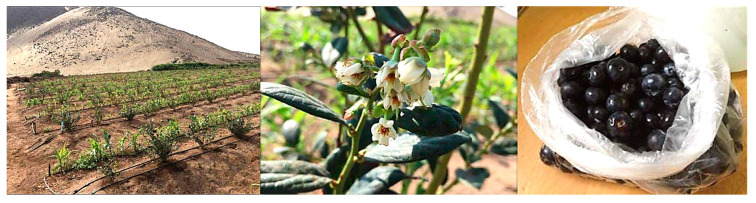
Leaves, flowers, and fruits of *Vaccinium corymbosum* L. collected in the district of Viru, Region of La Libertad, Peru.

**Table 1 plants-13-01643-t001:** Chemical composition of Peruvian *Vaccinium corymbosum* L. fruit extract determined by HPLC-ESI-QTOF-MS/MS.

N°	Proposed Compound	Molecular Formula	RT (min)	Mode of Ionization	Molecular Weight	Theoretical (*m*/*z*)	Observed (*m*/*z*)
**1**	Hydroxymethyl coumarin	C_10_H_8_O_3_	2.2	[M + H]^+^	176.047	177.0546	177.0505
**2**	Chlorogenic acid	C_16_H_18_O_9_	2.7	[M + H]^+^	354.095	355.1024	355.0957
**3**	p-Coumaroyl sinapoyl tartaric acid	C_24_H_22_O_12_	3.3	[M + H]^+^	502.428	503.1184	503.1082
**4**	5-O-Caffeoylshikimic acid	C_16_H_16_O_8_	4.1	[M + H]^+^	336.293	337.0918	337.0851
**5**	3,5-Di-O-caffeoylquinic acid	C_25_H_24_O_12_	4.3	[M + H]^+^	516.451	517.1341	517.1235
**6**	Myricetin-3-O-glucoside	C_21_H_20_O_13_	4.8	[M + H]^+^	480.376	481.0977	481.0875
**7**	3,5-Dicaffeoylquinic acid methyl ester	C_26_H_26_O_12_	5.3	[M + H]^+^	530.477	531.1497	531.1387
**8**	Caffeoyl coumaroylquinic acid	C_25_H_24_O_11_	5.5	[M + H]^+^	500.451	501.1391	501.1289
**9**	Quercetin-3-O-glucoside	C_21_H_20_O_12_	6.0	[M + H]^+^	464.376	465.1028	465.0931
**10**	Quercetin-3-O-arabinoside	C_20_H_18_O_11_	7.1	[M + H]^+^	434.350	435.0922	435.0822
**11**	Quercitrin	C_21_H_20_O_11_	7.9	[M + H]^+^	448.377	449.1078	449.0986

**Table 2 plants-13-01643-t002:** Free binding energy results (in kcal.mol^−1^) from molecular docking calculations of blueberry fruit extract with antioxidant proteins such as NADPH oxidase (2CDU), xanthine oxidase (3NRZ), and superoxide dismutase (4MCM) and antidepressant proteins like sodium/potassium-transporting ATPase alpha-1 chain (1MO7), monoamine oxidase (1O5W), and serotonin receptor 5-HT_2_ (6A94).

Compounds	Antioxidant			Antidepressant			Average	
	Free Binding Energy						Antioxidant	Antidepressant
	2CDU	3NRZ	4MCM	1MO7	1O5W	6A94		
**1**	−6.7	−7.0	−5.7	−5.5	−7.6	−6.8	−6.5	−6.6
**2**	−8.8	−8.3	−6.5	−6.6	−9.5	−8.4	−7.9	−8.2
**3**	−9.1	−8.3	−4.7	−6.7	−9.2	−9.2	−7.9	−8.4
**4**	−8.7	−8.6	−6.8	−6.8	−9.8	−8.7	−8.0	−8.4
**5**	−9.8	−9.9	−7.7	−6.9	−7.7	−9.4	−9.1	−8.0
**6**	−8.9	−8.6	−7.2	−7.5	−7.7	−8.9	−8.2	−8.0
**7**	−9.2	−9.1	−7.7	−7.1	−11.3	−9.1	−8.7	−9.2
**8**	−9.2	−8.8	−7.4	−6.1	−11.1	−11.0	−8.5	−9.4
**9**	−9.4	−8.9	−7.1	−7.6	−8.1	−8.7	−8.5	−8.1
**10**	−9.6	−9.3	−6.6	−7.6	−8.2	−9.2	−8.5	−8.3
**11**	−10.2	−9.4	−7.0	−7.9	−7.7	−8.6	−8.9	−8.1
Average	−9.0	−8.7	−7.0	−6.9	−8.9	−8.8	−8.2	−8.2

Values are listed with a three-colored scheme from red (high energy) to green (low energy).

**Table 3 plants-13-01643-t003:** ADMET properties of chemical constituents of Peruvian blueberry.

	Property									
	Absorption			Distribution			Metabolism	Excretion	Toxicity	
	Model Name									
Compounds	Caco-2	IA	SP	VDss	BBB	CNS	CYP2D6/ CYP3A4 Inhibitor	TC	Oral Rat Acute Tox. (LD_50_)	Oral Rat Chronic Tox. (LOAEL)
**1**	1.259	71.519	−2.374	−0.232	−0.160	−2.220	No/No	0.830	2.032	2.550
**2**	−0.644	13.558	−2.735	−0.291	−1.499	−3.973	No/No	0.353	2.238	3.341
**3**	−0.640	0.00	−2.735	0.146	−2.327	−4.075	No/No	−0.024	2.482	4.243
**4**	−0.627	30.806	−2.731	−0.578	−1.251	−3.684	No/No	0.439	2.068	2.833
**5**	−0.850	9.831	−2.735	0.780	−2.030	−3.991	No/No	0.025	3.072	4.084
**6**	−1.037	29.655	−2.735	0.056	−2.592	−4.731	No/No	0.486	2.551	4.118
**7**	−0.110	41.849	−2.735	0.740	−1.999	−3.957	No/No	0.073	3.009	3.352
**8**	−0.814	13.060	−2.735	0.048	−1.848	−3.854	No/No	0.206	2.637	2.717
**9**	−0.865	33.986	−2.735	0.275	−2.185	−4.658	No/No	0.541	2.634	4.097
**10**	−0.674	48.463	−2.735	0.204	−1.962	−4.378	No/No	0.549	2.711	3.620
**11**	−0.682	49.287	−2.735	0.212	−1.984	−4.319	No/No	0.549	2.715	3.583

Caco-2: Caucasian colon adenocarcinoma permeability (log Papp in 10^−6^ cm/s); IA: intestinal absorption (% absorbed); SP: skin permeability (log Kp); VDss: steady-state volume of distribution (log L/kg); BBB: blood–brain barrier permeability (log BB); CNS: central nervous system (log PS); CYP2D6: cytochrome P450 2D6 inhibitor; CYP3A4: cytochrome P450 3A4 inhibitor; TC: total clearance (log mL/min/kg); LD_50_: lethal dose, 50% (mol/Kg); LOAEL: lowest observed adverse effect level (log mg/kg bw/day).

## Data Availability

Data are contained within the article and Appendix A.

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
