# Peer review of "Chemical Composition, Antioxidant Activities, Antidepressant Effect, and Lipid Peroxidation of Peruvian Blueberry: Molecular Docking Studies on Targets Involved in Oxidative Stress and Depression"

_plants, 2024, doi:10.3390/plants13121643_

Round 1
Reviewer 1 Report
Comments and Suggestions for Authors
In this manuscript (plants-3015746) entitled "Chemical Composition, Antioxidant Activities, Antidepressant Effect, and Lipid Peroxidation of Peruvian Blueberry: Molecular Docking Studies on Targets Involved in Oxidative Stress and Depression" submitted to Plants, Iván M. Quispe-Díazy and colleagues have analyzed analyzed the components of the fruit of the Peruvian blueberry using liquid chromatography coupled with mass spectrometry HPLC–ESI–QTOF-MS/MS, and 11 compounds were identified. Furthermore, we assessed the in vitro antioxidant activities, antidepressant effects, protective effects on lipid peroxidation (in the serum, brain, liver, and stomach), molecular docking of the identified compounds with proteins involved in oxidative stress (NADPH oxidase, superoxide dismutase, and xanthine oxidase), and depression (serotonin receptor 5HT-2, monoamine oxidase, and ATPase). This research is complete and convincing, but some points need to be revised to improve the quality of this research.
Major points:
1. Authors should consider to remove Figure 1 from the revision, since this figure is a simple introduction.
2. For Figure 3, pictures to show the V. corymbosum fruits should be included in the revision..
3. Authors should consider to include a graph to summarize main conclusion of this study.
Minor points:
1. Authors need to standardize references according to the Plants template.
2. Full names of abbreviations like HPLC–ESI–QTOF-MS/MS and NADPH should be spelt out at their first appearance. Authors should check all abbreviations employed in the manuscript.
Author Response
Reviewer 1:
In this manuscript (plants-3015746) entitled "Chemical Composition, Antioxidant Activities, Antidepressant Effect, and Lipid Peroxidation of Peruvian Blueberry: Molecular Docking Studies on Targets Involved in Oxidative Stress and Depression" submitted to Plants, Iván M. Quispe-Díazy and colleagues have analyzed analyzed the components of the fruit of the Peruvian blueberry using liquid chromatography coupled with mass spectrometry HPLC–ESI–QTOF-MS/MS, and 11 compounds were identified. Furthermore, we assessed the in vitro antioxidant activities, antidepressant effects, protective effects on lipid peroxidation (in the serum, brain, liver, and stomach), molecular docking of the identified compounds with proteins involved in oxidative stress (NADPH oxidase, superoxide dismutase, and xanthine oxidase), and depression (serotonin receptor 5HT-2, monoamine oxidase, and ATPase). This research is complete and convincing, but some points need to be revised to improve the quality of this research.
We are grateful to the referee for the suggestions. In the revised version, we have incorporated their recommendations.
Major points:
1. Authors should consider to remove Figure 1 from the revision, since this figure is a simple introduction.
Answer: We agree with the reviewer’s comment. Both you and the other reviewer have commented on Figure 1. This figure has been deleted.
2. For Figure 3, pictures to show the V. corymbosumfruits should be included in the revision.
Answer: We agree. Pictures of V. corimbosum have been added to the plant material section.
3. Authors should consider to include a graph to summarize main conclusion of this study.
Answer: We appreciate the reviewer comments. In response, we included a graphical abstract to present the study's essential conclusions in a visually appealing and succinct manner.
Minor points:
1. Authors need to standardize references according to thePlants
Answer: The references were standardized according to the Plants journal
2. Full names of abbreviations like HPLC–ESI–QTOF-MS/MS and NADPH should be spelt out at their first appearance. Authors should check all abbreviations employed in the manuscript.
Answer: The full names of the abbreviations were spelled out at their first appearance, and all abbreviations used in the manuscript were checked.

Reviewer 2 Report
Comments and Suggestions for Authors
Figure 1 is not necessary, this is a research manuscript, not a review manuscript.
There is a lack of positive control for antioxidant experiments.
Is there a justification for using 2 doses for the antidepressant activity? Why did the authors select 10 and 20 mg/kg?
Figure 5 caption. What does VC mean?
Section 3.2 When was collected the plant material? Who did identify the plant material? The GPS coordinates of plant collection should be mentioned.
The authors are encouraged to perform an additional model for evaluating the antidepressant activity of the plant extract.
Do any of the compounds identified in the plant extract have reports on their antidepressant activity?
Conclusion
What does high antioxidant activity mean?
"A high antioxidant activity of the lyophilized V. corymbosum fruit extract was found as compared to values obtained with samples of V. corymbosum fruits from Italy and Portugal". This sentence is part of the discussion section, not the conclusion section.
Comments on the Quality of English Language
Grammar and spelling should be checked and corrected throughout the manuscript. The authors should show a certificate of grammar and spelling correction when submitting the corrected version of the manuscript.
Author Response
Reviewer 2
We thanks the referee for his/her nice comment.
1. Figure 1 is not necessary, this is a research manuscript, not a review manuscript.
Answer: We agree with the reviewer’s comment. Both you and the other reviewer have commented on Figure 1. This figure has been deleted.
2. There is a lack of positive control for antioxidant experiments.
Answer: We appreciate the reviewer’s comment. In the revised manuscript, we have included in the Figure 2, positive control (quercetin and ascorbic acid)
3. Is there a justification for using 2 doses for the antidepressant activity? Why did the authors select 10 and 20 mg/kg?
Answer: There have been scientific reports of Vaccinium virgatum (Brazil) that exhibits antidepressant effects at doses ranging from 40 to 120 mg/kg (shown in references). Based on this information, we conducted a preliminary assessment of the compound at doses of 10 mg/kg, 20 mg/kg, 40 mg/kg, and 80 mg/kg. We selected the lowest doses (10 and 20 mg/kg) that showed the most promising antidepressant results, as indicated in the study.
4. Figure 5 caption. What does VC mean?
Answer: VC mean is the abbreviation for Vaccinum corymbosum. We have standardized the abbreviations in the revised manuscript.
5. Section 3.2 When was collected the plant material? Who did identify the plant material? The GPS coordinates of plant collection should be mentioned.
Answer: We have included a paragraph that includes the suggestion regarding the section plant material.
6. The authors are encouraged to perform an additional model for evaluating the antidepressant activity of the plant extract.
Answer: We have disagreed with the referee’s suggestion. In future, we plan to implement other models of antidepressant activity.
7. Do any of the compounds identified in the plant extract have reports on their antidepressant activity?
Answer: We have no references of any reported antidepressant activity of some of the compounds identified in the blueberry fruit extract in this study.
8. Conclusion
What does high antioxidant activity mean?
"A high antioxidant activity of the lyophilized V. corymbosum fruit extract was found as compared to values obtained with samples of V. corymbosum fruits from Italy and Portugal". This sentence is part of the discussion section, not the conclusion section.
Answer: We agree with the criticism made by the referee. This sentence has been removed from the manuscript.
9. Grammar and spelling should be checked and corrected throughout the manuscript. The authors should show a certificate of grammar and spelling correction when submitting the corrected version of the manuscript.
Answer: Thank you very much for catching these glaring and confusing errors, which we have corrected and revised the manuscript.

Round 2
Reviewer 2 Report
Comments and Suggestions for Authors
The justification for using 10 and 20 mg/kg plant extract should be included in section 3.8.
The authors can not conclude that the plant extract showed antidepressant activity with a single in vivo model.
The authors should state in section 2 that no previous works report the antidepressant activity of the compounds identified in the plant extract.
Section 2.2. What does good antioxidant activity mean?
Comments on the Quality of English Languageno comments
Author Response
Reviewer 2 (round 2)
We thanks the referee for his/her nice comment.
1. The justification for using 10 and 20 mg/kg plant extract should be included in section 3.8.
Answer: We appreciate the reviewer comments. This justification has been added in Section 3.8.
2. The authors can not conclude that the plant extract showed antidepressant activity with a single in vivo model.
Answer: We thank the reviewer, and the conclusions have been modified accordingly.
3. The authors should state in section 2 that no previous works report the antidepressant activity of the compounds identified in the plant extract.
Answer: We appreciate the reviewer comments. In response, we included a sentence about this request.
4. Section 2.2. What does good antioxidant activity mean?
Answer: I agree that this expression could be a little confusing. We apologize for this. Good antioxidant activity is characterized by efficient scavenging of free radicals, thereby protecting the body from oxidative stress and its associated diseases. The sentence “good antioxidant activity” has been changed to “high antioxidant activity”.

Round 3
Reviewer 2 Report
Comments and Suggestions for Authors
The manuscript can be accepted for publication
Comments on the Quality of English Languageno comments
Author Response
Editor:
This is a very high quality paper that should be recommended for acceptance. However, there remain points that I believe are necessary to address.
Answer: We are grateful to the editor for the suggestions. We have incorporated these recommendations in the revised manuscript. The additions have been underlined in green.
1.- Line 27: the in vitro and in vivo.....
Answer: We agree. The in vitro and in vivo have been added.
2.- The introduction is too short with no clear rationale on why this study is of importance? What does the study add in light of previous in vitro and in vivo studies? Why Vaccinium corymbosum was selected? What about other Vaccinium spp? This should be clearly addressed at the end of section.
Answer: We appreciate the reviewer comments. In response, we have modified the introduction.
3.- Include reference/s in line 85 (Figure 1).
Answer: We have included a reference in Figure 1.
4.- Combining results and discussion in one section make no sense. The discussion provides insufficient interpretation of the results in my opinion. It is unclear what was found in the study and other studies that authors compared with. This is confusing. The discussion should be in a separate section and prepared by organizing information according to: the main findings and comparison of these findings with those reported in the literature; the weaknesses/limitations of the study and in relation to other studies, information about the present analyses and the implications of this study and future research directions.
Answer: We agree with the criticism made by the Editor. We have included a separate discussion of the results, including the requested suggestions.
5.- Authors should discuss why the dose ranged from 2.032–3.072 mol/kg indicates low toxicity, while different doses of blueberry fruit extract in other in vivo/vitro studies might be toxic.
Answer: We appreciate the editor´s comment. In the revised manuscript, we have included a discussion about the dose range from 2.032 to 3.072 mol/kg, which indicates low toxicity.
6.- The paper needs significant editing for language and writing quality.
Answer: Thank you very much for bringing about these glaring and confusing errors, which we have corrected and revised the manuscript.
